# ORDER MATTERS: IMPROVING DOMAIN ADAPTATION BY REORDERING DATA

## ABSTRACT

Domain shift remains a key challenge in deploying machine learning models to the real world. Unsupervised domain adaptation (UDA) aims to address this by minimising domain discrepancy during training, but the discrepancy estimates suffer from high variance in stochastic settings, which can stifle the theoretical benefits of the method. This paper proposes Optimal Reordering of Data for Error-Reduced Estimation of Discrepancy (ORDERED), a novel unbiased stochastic variance reduction technique which reduces the discrepancy estimation error by optimising the order in which the training data are sampled. We consider two specific domain discrepancy losses (correlation alignment and the maximum mean discrepancy), formulate their stochastic estimation error as a function of the data sampling order, and propose a practical optimisation algorithm. Our simulations demonstrate reduced variance compared to related methods, and experiments on a domain shift image classification benchmark show improved target domain accuracy.

## 1 INTRODUCTION

Machine learning models often underperform when the test data distribution differs from the training distribution, a phenomenon known as domain shift. Improving robustness to domain shift has been a longstanding goal in machine learning, and is crucial to the widespread deployment of AI (Gulrajani & Lopez-Paz, 2021; Koh et al., 2021).

Unsupervised domain adaptation (UDA) is a popular strategy to addressing this problem, in which the aim is to learn feature representations which are invariant across a source and target domain. This can be achieved by minimising a "domain discrepancy" term during training, which characterises the mismatch between the source and target feature distributions. This paper will consider two specific and notable examples: the correlation alignment (CORAL) loss, which measures distance between covariance matrices (Sun & Saenko, 2016); and the maximum mean discrepancy (MMD), which measures distance between kernel mean embeddings of the distributions (Tzeng et al., 2014; Long et al., 2015; Li et al., 2018).

Although theoretically well-grounded (Ben-David et al., 2006; 2010; Redko et al., 2022), a key limitation to these methods is that empirically estimating the discrepancy term is subject to extremely high levels of noise (i.e., the estimators have high variance). This is especially the case when the features are high-dimensional and the sample sizes are small (as when training via minibatch gradient descent), and can lead to unstable training, suboptimal adaptation, and thus poor target domain model performance. Indeed, a large body of work has reported finding these methods to have a negligible or even negative impact on training compared to vanilla empirical risk minimisation (ERM) (Dubey et al., 2021; Gao et al., 2023; Gulrajani & Lopez-Paz, 2021; Koh et al., 2021; Napoli & White, 2023; 2024; Wang et al., 2019).

The estimator noise can be lowered through the use of variance reduction, and this has previously been shown to improve performance in the UDA setting (Napoli & White, 2024; Anonymous, 2025). Although a large number of such techniques exist, many require the loss to be additive over individual training examples, which renders them incompatible with UDA losses (which fundamentally depend on the interrelation between training examples). We defer to Anonymous (2025); Gower et al. (2020) for a full review of these techniques.

Our approach builds on Anonymous (2025), who reduce the variance via stratified sampling (Zhao & Zhang, 2014; Liu et al., 2020): the data are stratified using discrepancy-specific clustering objectives, and minibatches are formed by drawing a single instance uniformly and independently at random from each stratum. Weighted loss functions are then used to correct for imbalanced stratum sizes and ensure the losses remain unbiased.

This approach has three main shortcomings: 1) the strata are formed by clustering based on a surrogate objective, which does not always directly correspond to the estimator variance; 2) the strata are sampled independently, which limits the degree of variance reduction which can be achieved; 3) if the stratum sizes are highly imbalanced, convergence will be slow since it will take more training iterations to "see" all the examples in the larger strata.

To address Shortcomings 1 and 2, our paper proposes an additional step which directly and jointly optimises the sampling order of the data in each stratum. This step minimises a new surrogate objective closer to the true estimator variance. We call this method Optimal Reordering of Data for Error-Reduced Estimation of Discrepancy (ORDERED). To address Shortcoming 3, we also slightly amend the clustering algorithm to enforce a minimum cluster size.

Modification of the training data sampling order is a common area of research, though not normally with the specific goal of reducing variance. For example, curriculum learning (Bengio et al., 2009) is a well-known paradigm which presents examples in increasing order of difficulty, and a large literature of derived work exists (Wang et al., 2020). The ordering of priming prompts for large language models has also been shown to significantly affect performance; prior works have proposed genetic algorithms (Kumar & Talukdar, 2021) or entropy-based metrics (Lu et al., 2022) to find the optimal permutation. Relatedly, the training distribution can also be varied using a weight schedule to mix multi-domain data (Rukhovich et al., 2024). However, to our knowledge, our work is the first to choose the sampling order by explicitly solving a permutation problem with respect to variance, and certainly the first to do so in the context of UDA losses.

In the following sections, we introduce UDA variance reduction via stratified sampling, and propose a modified clustering algorithm with cluster size constraints. We then formulate the stochastic estimation errors of the MMD and CORAL losses as a function of the data order, and propose a practical optimisation algorithm. We conduct analyses of all novel elements using Monte Carlo simulations, and demonstrate the superiority of our method on a high-quality domain shift image classification benchmark.

## 2 METHOD

### 2.1 PRELIMINARIES

Given labelled source examples $x_{s,i}, y_{s,i}$ indexed by $i \in \mathcal{I}_s = \{1, \ldots, n_s\}$, and unlabelled target examples $x_{t,j}$ indexed by $j \in \mathcal{I}_t = \{1, \ldots, n_t\}$, the goal of UDA is to learn a model $h$ that minimises some task loss $L_{\text{task}}$ on the target domain. It is assumed $h$ decomposes into a featuriser $f$ and prediction head $g$, such that $h = g \circ f$. Since the target data are unlabelled, UDA methods instead minimise $L_{\text{task}}$ on the source domain, alongside a domain discrepancy loss $L_{\text{disc}}$ which aligns the source and target feature distributions:

$$\min_h \mathbb{E}\left[L_{\text{task}}\left(h\left(x_s\right), y_s\right) + \lambda L_{\text{disc}}\left(f\left(x_s\right), f\left(x_t\right)\right)\right], \tag{1}$$

where $\lambda \in \mathbb{R}^+$ controls the trade-off between the task and domain alignment objectives. This paper considers two specific options for $L_{\text{disc}}$, the MMD and CORAL. The MMD is defined as

$$L_{\text{MMD}}\left(f\left(x_s\right), f\left(x_t\right)\right) = \left\|\mathbb{E}\left[\phi\left(f\left(x_s\right)\right)\right] - \mathbb{E}\left[\phi\left(f\left(x_t\right)\right)\right]\right\|_{\mathcal{H}}^2 \tag{2}$$

where $\mathcal{H}$ is a reproducing kernel Hilbert space, and $\phi : \mathcal{Z} \to \mathcal{H}$ is an implicit mapping. $\mathcal{H}$ is associated with a unique positive-definite kernel $\kappa : \mathcal{Z} \times \mathcal{Z} \to \mathbb{R}$ for which the reproducing property $\kappa(z, z') = \langle \phi(z), \phi(z') \rangle_{\mathcal{H}}$ is satisfied. On the other hand, CORAL aims to minimise the (squared) Frobenius distance between the source and target feature covariance matrices:

$$L_{\text{CORAL}}\left(f\left(x_s\right), f\left(x_t\right)\right) = \left\|\text{Cov}\left[f\left(x_s\right)\right] - \text{Cov}\left[f\left(x_t\right)\right]\right\|_F^2. \tag{3}$$

At training iteration $m$, we select index subsets $B_s^{(m)} \subseteq \mathcal{I}_s$ and $B_t^{(m)} \subseteq \mathcal{I}_t$, each of cardinality $k$, and construct minibatches $\mathcal{B}_s^{(m)} = \left\{(x_{s,i}, y_{s,i}) \mid i \in B_s^{(m)}\right\}$ and $\mathcal{B}_t^{(m)} = \left\{x_{t,j} \mid j \in B_t^{(m)}\right\}$.

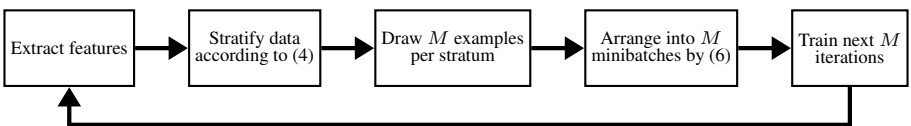

Figure 1: ORDERED training pipeline.

These are then used to compute stochastic losses $\widehat{L}_{\text{task}}^{(m)}$ and $\widehat{L}_{\text{disc}}^{(m)}$, and update $h$. Our aim is to reduce the discrepancy estimation error $\sum_m \left( \widehat{L}_{\text{disc}}^{(m)} - L_{\text{disc}}^{(m)} \right)^2$ over the course of the training, by optimising how $B_s^{(m)}$ and $B_t^{(m)}$ are chosen.

## 2.2 METHOD OVERVIEW

Unfortunately, $\widehat{L}_{\text{disc}}^{(m)}$ and $L_{\text{disc}}^{(m)}$ depend on the features at iteration $m$, making it hard to optimise them directly. However, they can be well-approximated using features from previous iterations, so long as the loss surface is locally smooth and the learning rate is sufficiently small (Liu et al., 2020). Intuitively, it can be assumed that features that are close to each other at iteration $m$ will still tend to be close at iteration $m + 1$. Therefore, future minibatches are predetermined in sets of $M$, based on features $z_{s,i} = f\left(x_{s,i}\right),\ z_{t,j} = f\left(x_{t,j}\right)$ extracted at the current training iteration.

We build ORDERED on top of stratified sampling (Anonymous, 2025). That is, we first partition $\mathcal{I}_s$ and $\mathcal{I}_t$ each into $k$ strata, $S_1, \ldots, S_k$ and $T_1, \ldots, T_k$ respectively. We then sample $M$-tuples $\widetilde{S}_h, \widetilde{T}_h$ uniformly at random from each stratum, which will form the next $M$ source and target minibatches. Specifically, the $m^{\text{th}}$ minibatches are defined as $B_s^{(m)} = \bigcup_h \widetilde{S}_h^{(m)}$, $B_t^{(m)} = \bigcup_h \widetilde{T}_h^{(m)}$, comprising the $m^{\text{th}}$ element from each tuple, and the tuple orderings jointly minimise a surrogate discrepancy estimation error based on $z_{s,i},\ z_{t,j}$. This approach ensures that the losses over the whole training remain unbiased. The overall training pipeline is shown in Figure 1.

## 2.3 STRATIFICATION

We construct the strata using dynamically-weighted kernel k-means clustering (Anonymous, 2025). To address Shortcoming 3, we add minimum cluster size constraints – this sacrifices some variance reduction in return for faster convergence during training. This section describes the clustering for $\mathcal{I}_s$; the same procedure can be repeated analogously for $\mathcal{I}_t$. For the MMD, the clustering objective is

$$\arg \min_{S_1, \ldots, S_k} \sum_{h=1}^{k} |S_h| \sum_{i \in S_h} \left\| \phi\left(z_{s,i}\right) - \frac{1}{|S_h|} \sum_{i \in S_h} \phi\left(z_{s,i}\right) \right\|_{\mathcal{H}}^2 \tag{4}$$

subject to $|S_h| \geq n_{\min}$. For CORAL, the objective is of the same form, but uses the specific mapping $\phi_c(z) = (z - \overline{z})(z - \overline{z})^T$. These objectives are derived from the variance expressions of $\widehat{L}_{\text{disc}}^{(m)}$, and are shown to be good surrogates for minimising the true variances when the data are sampled independently for each stratum and iteration (Anonymous, 2025).

(4) can be solved in a similar manner to Anonymous (2025), using a Lloyd's-style alternating optimisation algorithm (Lloyd, 1982). Specifically, the algorithm alternates between 2 steps:

1. **Distance Update:** Compute the distance matrix $P \in \mathbb{R}^{n_s \times k}$ from each datapoint to the centroid of each cluster using the kernel trick.

2. **Dynamically Weighted Assignment:** Compute the one-hot cluster assignment matrix $U \in \{0,1\}^{n_s \times k}$ that assigns each point to exactly one of the $k$ clusters.

$U$ is the solution to the quadratic program

$$\arg\min_U \sum_{i,h} \left[ U_{ih} P_{ih} \sum_i U_{ih} \right] \tag{5}$$

$$\text{subject to} \quad 0 \leq U_{ih} \leq 1, \ \sum_h U_{ih} = 1, \ \sum_i U_{ih} \geq n_{\min}.$$

Since the Hessian of (5) is indefinite in general, this problem is nonconvex and thus finding the global minimum is NP-hard. Although the problem as currently defined could be readily input to a gradient-based interior point method (to find a local minimum), these have $O\left((kn_s)^3\right)$ complexity, and are impractical above a few hundred data points. Instead, we solve (5) using a greedy heuristic in a similar manner to Anonymous (2025), but with an extra condition to satisfy the cluster size constraints. The algorithm constructs $U$ incrementally row-by-row, weighting the clusters using interim cluster size values. Indices are assigned freely while there are sufficient remaining datapoints to satisfy the constraints, after which point the possible allocations are restricted to clusters that do not yet reach the minimum size. This algorithm runs in $O\left(kn_s\right)$ time, and is listed in Algorithm 1, where $R(x) = \begin{cases} x, \ x \geq 0; \\ 0, \ x < 0 \end{cases}$ is the ramp function.

---

**Algorithm 1** Constrained weighted cluster assignments

---

**Require:** $P \in \mathbb{R}^{n_s \times k}$, $n_{\min} \in \mathbb{N}^+$
**Ensure:** $U \in \{0,1\}^{n_s \times k}$, $\sum_h U_{ih} = 1$
1: $U \leftarrow 0_{n_s \times k}$
2: $n_1, \ldots, n_k \leftarrow 0$   ▷ Interim cluster sizes
3: $r \leftarrow n_s$   ▷ Number of remaining assignments
4: **for all** $i \in \{1, \ldots, n_s\}$ **do**
5:   $H \leftarrow \left\{ h \in \{1, \ldots, k\} : n_h < n_{\min} \text{ or } r \geq \sum_{h=1}^k R\left(n_{\min} - n_h\right) \right\}$
6:   $h \leftarrow \arg\min_{h \in H} P_{ih}\left(n_h + 1\right)$
7:   $U_{ih} \leftarrow 1$
8:   $n_h \leftarrow n_h + 1$
9:   $r \leftarrow r - 1$
10: **end for**
11: **return** $U$

---

Figure 2 shows how the loss attained by minimising (5) is affected by the hyperparameter $n_{\min}$. The input comprises Euclidean distances between samples from a 2D standard normal distribution. We use a small problem with $n_s = 200$ and $k = 5$, which allows us to compare Algorithm 1 with a commercial interior point solver (The MathWorks Inc., 2021). We also test an unweighted constrained assignment, which is a linear problem and can thus be solved quickly using linear programming, but does not optimise the same objective. As expected, increasing $n_{\min}$ restricts the feasible problem space, which tends to increase the achievable loss; however, the greedy algorithm appears less affected by this than the interior point method. Note that as $n_{\min}$ approaches $n_s/k$, the clusters tend to equal sizes, which is why the unweighted optimiser approaches the weighted optimisers at this point.

## 2.4 OPTIMISING SAMPLE ORDER

First, we present the sampling order optimisation problem in canonical integer programming form. To proceed, let $\alpha \in \{0,1\}^{n_s \times M}, \beta \in \{0,1\}^{n_t \times M}$ be binary indicator variables for the source and target indices respectively, such that $\alpha_{im} = \begin{cases} 1, \ i \in B_s^{(m)}; \\ 0, \text{ otherwise.} \end{cases}$, and likewise for $\beta$. Define also cluster size vectors $\mathbb{S} \in \mathbb{N}^{n_s}, \mathbb{T} \in \mathbb{N}^{n_t}$, where $\mathbb{S}_i = |S_h| \Leftrightarrow i \in S_h$ (i.e., $\mathbb{S}_i$ is the size of the cluster containing index $i$), and equivalently for $\mathbb{T}_j$, used to weight the distance estimates to correct the sampling bias introduced by the imbalanced clusters. Finally, let $\widetilde{B}_s = \bigcup_h \widetilde{S}_h = \bigcup_m B_s^{(m)}$ and

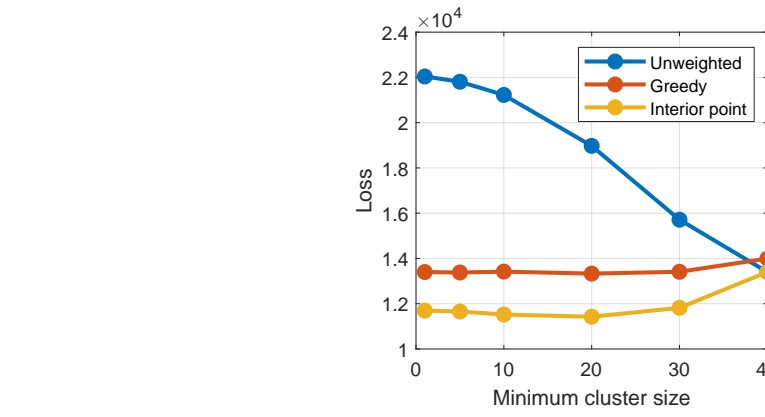

Figure 2: Objective value of (5) vs minimum cluster size $n_{\min}$ for three different optimisation algorithms.

$\widetilde{B}_t = \bigcup_h \widetilde{T}_h = \bigcup_m B_t^{(m)}$ be the union of all source and target indices for the next $M$ minibatches. The optimisation problem is thus

$$\min_{\alpha,\beta} \sum_{m=1}^M \left( \widehat{D}^{(m)} - D_0 \right)^2 \tag{6}$$

$$\text{subject to} \sum_m \alpha_{im} = 1, i \in \widetilde{B}_s, \sum_m \beta_{jm} = 1, j \in \widetilde{B}_t \tag{7}$$

$$\sum_m \alpha_{im} = 0, i \notin \widetilde{B}_s, \sum_m \beta_{jm} = 0, j \notin \widetilde{B}_t \tag{8}$$

$$\sum_{i \in S_h} \alpha_{im} = 1, \sum_{j \in T_h} \beta_{jm} = 1, \tag{9}$$

$$\alpha_{im}, \beta_{jm} \in \{0, 1\}, \tag{10}$$

where $D_0$ is the surrogate "reference" discrepancy over the full dataset (approximating $L_{\text{disc}}^{(m)}$), and $\widehat{D}^{(m)}$ expresses the stochastic losses in terms of $\alpha$ and $\beta$ (approximating $\widehat{L}_{\text{disc}}^{(m)}$). The reference (squared) MMD is given by

$$D_{0,\text{MMD}} = \left\| \frac{1}{n_s} \sum_{i=1}^{n_s} \phi\left( z_{s,i} \right) - \frac{1}{n_t} \sum_{j=1}^{n_t} \phi\left( z_{t,j} \right) \right\|_{\mathcal{H}}^2, \tag{11}$$

and the stochastic estimates are

$$\widehat{D}_{\text{MMD}}^{(m)} = \left\| \frac{1}{n_s} \sum_{i=1}^{n_s} \alpha_{im} \mathbb{S}_i \phi\left( z_{s,i} \right) - \frac{1}{n_t} \sum_{j=1}^{n_t} \beta_{jm} \mathbb{T}_j \phi\left( z_{t,j} \right) \right\|_{\mathcal{H}}^2, \tag{12}$$

or, in terms of kernel evaluations,

$$\widehat{D}_{\text{MMD}}^{(m)} = \frac{1}{n_s^2} \sum_{i,i'=1}^{n_s} \alpha_{im} \alpha_{i'm} \mathbb{S}_i \mathbb{S}_{i'} \kappa\left( z_{s,i}, z_{s,i'} \right) + \frac{1}{n_t^2} \sum_{j,j'=1}^{n_t} \beta_{jm} \beta_{j'm} \mathbb{T}_j \mathbb{T}_{j'} \kappa\left( z_{t,j}, z_{t,j'} \right)$$

$$- \frac{2}{n_s n_t} \sum_{i,j=1}^{n_s,n_t} \alpha_{im} \beta_{jm} \mathbb{S}_i \mathbb{T}_j \kappa\left( z_{s,i}, z_{t,j} \right). \tag{13}$$

The reference CORAL loss is

$$D_{0,\text{CORAL}} = \left\| C_{s,0} - C_{t,0} \right\|_F^2, \tag{14}$$

where $C_{s,0}$ and $C_{t,0}$ are the sample covariance matrices of $z_s$ and $z_t$ respectively. The stochastic estimates are

$$\widehat{D}_{\text{CORAL}}^{(m)} = \left\| \widehat{C}_s^{(m)} - \widehat{C}_t^{(m)} \right\|_F^2$$

$$\widehat{C}_s^{(m)} = \frac{1}{n_s - 1} \sum_{i=1}^{n_s} \alpha_{im} \mathbb{S}_i \left( z_{s,i} - \widehat{\mu}_s^{(m)} \right) \left( z_{s,i} - \widehat{\mu}_s^{(m)} \right)^T$$

$$\widehat{C}_t^{(m)} = \frac{1}{n_t - 1} \sum_{j=1}^{n_t} \beta_{jm} \mathbb{T}_j \left( z_{t,j} - \widehat{\mu}_t^{(m)} \right) \left( z_{t,j} - \widehat{\mu}_t^{(m)} \right)^T \tag{15}$$

$$\widehat{\mu}_s^{(m)} = \frac{1}{n_s} \sum_{i=1}^{n_s} \alpha_{im} \mathbb{S}_i z_{s,i}, \quad \widehat{\mu}_t^{(m)} = \frac{1}{n_t} \sum_{j=1}^{n_t} \beta_{jm} \mathbb{T}_j z_{t,j}.$$

Note that the MMD objective is a quartic matrix polynomial in $\alpha$ and $\beta$, whereas the CORAL objective is of order 8. However, since $\alpha$ and $\beta$ are binary variables, the problem can be linearised via standard methods (Balas & Mazzola, 1984). At this point, the problem could be input as-is into a standard integer programming solver. However, this will not be practical for large datasets due to the size of the problem. Instead, by considering the specific structure of the problem, we propose a faster heuristic which searches for a local minimum using a greedy strategy.

The approach begins with an initial random data order and reduces the objective by iteratively swapping pairs of indices. Specifically, the algorithm executes a single pass through the data, choosing the optimal swap out of the *remaining* elements in the same stratum via exhaustive search. This means $\frac{M(M-1)}{2}$ objective comparisons are performed per stratum, and thus $kM(M-1)$ comparisons in total (for both $\widetilde{B}_s$ and $\widetilde{B}_t$). This algorithm is guaranteed to find a permutation at least as good as the initial permutation. The algorithm is listed fully in Algorithm 2.

---

**Algorithm 2** ORDERED

1: Initialise each $M$-tuple $\widetilde{S}_1, \widetilde{T}_1, ..., \widetilde{S}_k, \widetilde{T}_k$ with a random permutation
2: **for all** $m \in \{1, \ldots, M\}$ **do**     ▷ Iteration index
3:     **for all** $h \in \{1, \ldots, k\}$ **do**     ▷ Stratum index
4:         Swap elements $\widetilde{S}_h^{(m)}$ and $\widetilde{S}_h^{(m_s)}$, where $m_s \in \{m, \ldots, M\}$ and minimises (6).
5:         Swap elements $\widetilde{T}_h^{(m)}$ and $\widetilde{T}_h^{(m_t)}$, where $m_t \in \{m, \ldots, M\}$ and minimises (6).
6:     **end for**
7: **end for**
8: **return** $\widetilde{S}_1, \widetilde{T}_1, ..., \widetilde{S}_k, \widetilde{T}_k$

---

We use Monte Carlo simulations to analyse the performance characteristics of Algorithm 2 with respect to the parameters $k$ and $M$. Specifically, we compute the variance of stochastic MMD estimates using a linear kernel (that is, estimating the squared Euclidean distance between distribution means) between a source and target dataset comprising 2D standard normal data with $n_s = n_t = 4,000$. Figure 3a compares the variance across different values of $k$ for 3 samplers: uniform random sampling, stratified sampling (Anonymous, 2025), and ORDERED, with $M = 100$. ORDERED achieves up to 2 orders of magnitude reduction in variance compared to stratified sampling, and 4 orders of magnitude reduction compared to uniform random sampling.

Figure 3b shows how the variance changes with $M$, with $k = 20$. As expected, the variance reduces significantly at first, since the optimisation has greater degrees of freedom. However, perhaps counter-intuitively, it can be seen to increase again for $M > 50$. We posit that this is because the smaller problem size induces more noise, which helps to avoid local minima and achieve a better global solution. Furthermore, for lower $M$, the surrogate objective being optimised $\left( \widehat{D}^{(m)} - D_0 \right)^2$ is closer on average to the true deviation $\left( \widehat{L}_{\text{disc}}^{(m)} - L_{\text{disc}}^{(m)} \right)^2$, which also improves reduction in variance. As well as the solution quality, $M$ is a trade-off in computational cost: lower $M$ requires more frequent extraction of features, but higher $M$ increases the complexity of Algorithm 2 quadratically.

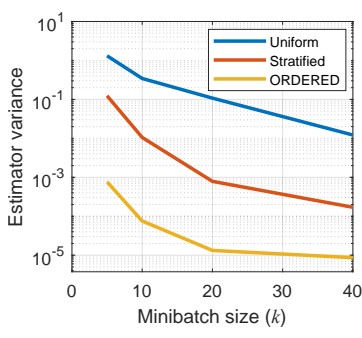
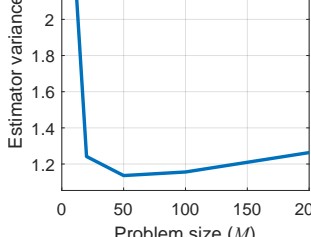

(a) Estimator variance vs minibatch size for ORDERED and two ablations.

(b) Estimator variance vs $M$ for ORDERED.

Figure 3: The performance characteristics of Algorithm 2.

Thus, the choice of $M$ is influenced by a complex combination of factors. For simplicity, we choose to fix $M = 100$ for the remainder of the experiments, which is the same update frequency chosen by Anonymous (2025), and based on empirical observations from previous work (Liu et al., 2020).

## 3 EXPERIMENTS

In this section, the proposed method is evaluated in realistic training conditions, to assess whether the observed reduction in variance translates to an increase in test accuracy. Experiments are conducted using the DomainBed framework (Gulrajani & Lopez-Paz, 2021) on the Spawrious domain shift benchmark (Lynch et al., 2023). The task comprises classifying images of dogs into four breeds, across six domains characterised by the background environments (desert, jungle, snow etc.). The images are synthetically generated, which allows for controlled introduction of spurious correlations, and results in a higher-quality benchmark than earlier options. A random subset of 18,664 images from the full dataset is used to speed up testing. The benchmark defines six training-evaluation splits, covering two spurious correlation types (One-to-One (O2O) and Many-to-Many (M2M)) and three difficulty levels (Easy, Medium, Hard).

The domain discrepancies are measured between the union of all training data and a held-out subset of the evaluation set. For the MMD, we use a radial basis function (RBF) mixture kernel (Li et al., 2018), given by $\kappa(z, z') = \sum_{\gamma \in \mathcal{G}} e^{-\gamma \|z - z'\|^2}$ with $\mathcal{G} = \{0.001, 0.01, 0.1, 1, 10\}$. For the clustering, we set $n_{\min} = M = 100$, and sample $\widetilde{S}_h, \widetilde{T}_h$ without replacement, which provides a further reduction in variance (Gower et al., 2020).

The model comprises a pretrained ResNet-18 architecture (He et al., 2015), which is finetuned on the training data using the Adam optimiser (Kingma & Ba, 2014) for 3,000 iterations. Hyperparameters are tuned with a random search of size 10 using an in-distribution (training domain) validation set, independently for each sampler. The entire set of experiments is repeated 3 times for reproducibility, using different random seeds for hyperparameters, weight initialisations, and dataset splits. All other hyperparameter choices and training details follow the DomainBed default options.

In total, 3 sampling methods are compared. These are: uniform random sampling; stratified sampling (Anonymous, 2025); and ORDERED. Table 1 shows the average test accuracy and standard errors over the 3 repeats for each of the 6 data splits, for both the CORAL and MMD algorithms. The results confirm the importance of effective variance reduction when estimating UDA losses. Compared to uniform random sampling, ORDERED increases average accuracy by 7.5 and 13.4 percentage points for CORAL and MMD respectively, and by 2.1 and 3.7 percentage points compared to stratified sampling. The performance gains are consistently higher for the MMD than for CORAL (note that the average accuracy without variance reduction is the same for both). We posit that this is because the MMD estimates are noisier due to their incorporation of higher-order statistics, making the benefits of variance reduction more pronounced. Overall, there is no clear relationship between the type or difficulty of the data split, and the magnitude of accuracy improvement.

Table 1: Average test accuracy for each data split and training algorithm.

(a) CORAL

| Sampler | O2O-Easy | O2O-Medium | O2O-Hard | M2M-Easy | M2M-Medium | M2M-Hard | Average |
|---|---|---|---|---|---|---|---|
| Uniform | 69.4 ± 3.6 | 56.0 ± 2.0 | 64.9 ± 0.6 | 79.1 ± 2.3 | 54.4 ± 2.0 | 48.3 ± 0.8 | 62.0 ± 0.9 |
| Stratified | 83.4 ± 7.1 | **61.9 ± 1.6** | 71.2 ± 8.4 | **85.2 ± 3.6** | 59.4 ± 1.6 | 49.4 ± 2.4 | 68.4 ± 2.0 |
| ORDERED | **88.2 ± 2.2** | 61.6 ± 1.6 | **78.1 ± 3.5** | 84.1 ± 5.0 | **60.5 ± 2.6** | **50.5 ± 2.1** | **70.5 ± 1.3** |

(b) MMD

| Sampler | O2O-Easy | O2O-Medium | O2O-Hard | M2M-Easy | M2M-Medium | M2M-Hard | Average |
|---|---|---|---|---|---|---|---|
| Uniform | 73.8 ± 2.2 | **61.9 ± 1.9** | 60.5 ± 1.9 | 80.5 ± 4.0 | 51.3 ± 2.4 | 48.1 ± 0.7 | 62.7 ± 1.0 |
| Stratified | 91.7 ± 2.8 | 60.8 ± 0.4 | 83.4 ± 3.7 | 84.2 ± 1.2 | 60.0 ± 11.3 | 54.1 ± 4.8 | 72.4 ± 2.2 |
| ORDERED | **93.5 ± 1.3** | 56.4 ± 2.4 | **85.1 ± 1.9** | **88.6 ± 0.8** | **70.5 ± 7.8** | **62.1 ± 10.9** | **76.1 ± 2.3** |

## 4 CONCLUSION

This paper introduced ORDERED, a novel stochastic variance reduction method for UDA based on reordering the training data. We showed that the training data sampling order drastically influences the stochastic estimation error of the MMD and CORAL losses, which in turn significantly affects target domain performance. To address this, we formulated the estimation error as a function of the data order, and proposed a practical optimisation algorithm.

We believe the most promising direction for future work is in improving the optimisation procedure, for instance by applying metaheuristics such as simulated annealing or tabu search to enhance robustness against local minima. The approach could also be extended to other UDA objectives or a domain generalisation setting.

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
