# OpenReview forum: "Order Matters: Improving Domain Adaptation by Reordering Data"
_ICLR.cc/2026/Conference — ICLR 2026 Conference Withdrawn Submission_

### Official Review · Reviewer_tsd2 · 2025-10-24

**Soundness:** 2
**Presentation:** 1
**Contribution:** 1
**Rating:** 2
**Confidence:** 4

**Summary:**

Discrepancy-based domain adaptation methods, such as MMD and CORAL, usually suffer from the high variance of the discrepancy loss due to the randomness of the mini-batch. This work proposes to optimize the sampling order to reduce the discrepancy estimation error. In the experiments, the proposed method works better than the base stratified sampling method.

**Strengths:**

1. This paper proposes an improved domain adaptation method based on the author’s previous work, and there is a degree of novelty in the proposed method.
2. The effectiveness of the proposed method is confirmed through the experiments.

**Weaknesses:**

1. The contribution of this work is very narrow. This paper proposes a variant method of the author’s previous work, but it does not demonstrate the applicability of the proposed method to other existing works. In addition, I do not find any reason to publish it as an independent paper separate from the authors' base paper.
2. Experimental validation is not enough. This paper does not provide a comparison with the other existing domain adaptation methods, and the superiority of the proposed method is unclear.
3. There are not enough analyses of the proposed method given. Even an ablation study has not been conducted, and the validity of each proposed component is unclear.
4. Discussion with the related works is not given. A separate related work section is not provided, and the relationship between the other domain adaptation methods and this work is unclear.
5. The paper presentation is not good and is difficult to read. The description of this paper heavily depends on the author’s previous work, and we cannot understand this work only from this paper.

**Questions:**

1. To what extent are the findings of this study compatible with other existing domain adaptation works?

---

### Official Review · Reviewer_fFfY · 2025-10-26

**Soundness:** 2
**Presentation:** 1
**Contribution:** 2
**Rating:** 2
**Confidence:** 5

**Summary:**

This paper addresses a key limitation of Unsupervised Domain Adaptation (UDA): high variance in stochastic estimates of domain discrepancy losses (specifically Correlation Alignment (CORAL) and Maximum Mean Discrepancy (MMD)), which undermines training stability and target domain performance. The authors propose ORDERED, a novel unbiased variance reduction technique that optimizes the sampling order of training data to minimize discrepancy estimation error.

**Strengths:**

1. The paper focuses on a well-documented but underaddressed limitation of UDA (high variance in discrepancy estimators) that directly harms real-world deployment.
2. This paper is simple and easy to follow.

**Weaknesses:**

1. The paper does not ablate the impact of individual components (e.g., constrained clustering vs. order optimization). For example, would ordered sampling alone (without constrained clustering) yield similar gains? This limits understanding of ORDERED’s critical parts.
2. Lack of novelty. The proposed method is a simple combination of MMD and CORAL for evaluating the discrepancy.
3. Limited Benchmark Coverage. Generalization to other UDA scenarios (e.g., real-world datasets like Office-Home, VisDA, or non-image domains like text) is unproven.

**Questions:**

The "stratified sampling" baseline (Anonymous, 2025) is cited but not fully described. For example, does it use the same clustering objective or minibatch size as ORDERED? Without this, it is hard to isolate whether gains come from ordering or other modifications (e.g., constrained clustering).

---

### Official Review · Reviewer_xNvA · 2025-10-31

**Soundness:** 2
**Presentation:** 2
**Contribution:** 2
**Rating:** 2
**Confidence:** 5

**Summary:**

This paper proposes ORDERED, a variance reduction scheme for unsupervised domain adaptation (UDA) that treats the within stratum order of samples as an explicit degree of freedom on top of stratified sampling. The method builds two components: (i) a stratification step via dynamically weighted (kernel) k means with a minimum cluster size constraint, and (ii) a greedy in cluster swap procedure over the next $M$ minibatches to minimize the deviation between the batch level discrepancy $\widehat{D}^{(m)}$ (for MMD or CORAL) and a global reference $D_0$. Simulation and DomainBed style experiments (Spawrious) show reduced variance and accuracy gains relative to uniform/stratified sampling. The idea of order aware variance control for non additive losses is interesting, but the paper needs stronger support on unbiasedness conditions, experimental breadth and fairness, complexity/overhead accounting, and reproducibility before being ready.

**Strengths:**

The paper offers a simple, original idea: treat the within-batch ordering of stratified samples as a knob to reduce the variance of non-additive discrepancy losses like MMD and CORAL. This shifts attention from changing models or losses to shaping the sampling process, which is a fresh and intuitive perspective for domain adaptation.

The method is practical and easy to plug into standard UDA training. It adds a lightweight in-cluster swap routine on top of familiar backbones and losses, keeps hyperparameters understandable, and remains broadly compatible with existing pipelines.

Empirically, the paper shows consistent gains over uniform and plain stratified sampling and clear variance reductions in controlled simulations. The analyses around the number of strata and update frequency help explain when the effect is strongest, and the visualizations make the mechanism easy to follow.

**Weaknesses:**

Table 1 compares samplers (Uniform/Stratified/ORDERED) under CORAL or MMD training, but it does not specify the MMD estimator used (biased V stat vs unbiased U stat). Estimator choice is orthogonal to the sampler: it changes the variance/stability of the minibatch discrepancy that method optimizes against. Without stating (i) which estimator is used, (ii) whether both $D_0$ and $\widehat{D}^{(m)}$ use the same estimator, and (iii) whether all samplers are evaluated with the same estimator, part of the reported gains could stem from the estimator rather than the ordering strategy.


Results are reported mainly on the synthetic Spawrious setup; at least one real cross domain benchmark is needed. It is also ambiguous whether the reference $D_0$ is computed using samples from the evaluation/test split, which would constitute leakage.


The paper discusses high-order objectives and mentions linearization, but provides no quantitative accounting of variables/constraints or the runtime/memory overhead of the greedy ordering.

The paper claims that the weighted batch discrepancies are unbiased. However, the implemented protocol, which uses stratified sampling with one draw per stratum without replacement and additionally applies greedy in-cluster swaps, alters inclusion probabilities and induces dependencies among pairs in non-additive statistics (MMD/CORAL). Unbiasedness must therefore be established for this exact sampler, yet the submission provides no conditions or proof supporting that claim.

The figure only shows the simulated effects of $k$ and $M$, lacking a systematic ablation and sensitivity analysis on final accuracy; reproducibility is further limited by missing training details (batch size, LR schedule, weight decay).

The paper states an+7.5 pp for ORDERED over Uniform under CORAL, but Table 1 shows 70.5 − 62.0 = +8.5 pp. Unify the overloaded k (clusters vs. batch)

The entire text is only validated on ResNet-18. The benefits of ORDERED come from the difference estimation on small batches, and its stability and magnitude may vary with the encoder capacity, structure (CNN vs ViT), and normalization method. It is recommended to supplement at least two different inductive biases/capacities of backbones (such as ResNet-50 and ViT-B/16 or ConvNeXt-T).

**Questions:**

1. Which MMD estimator do you use (biased V stat or unbiased U stat)? Do both $D_0$ and $\widehat{D}^{(m)}$ use the same estimator, and is the choice consistent across all samplers? Please add a minimal V vs U ablation to show the effect on variance and accuracy.

2. Is any evaluation/test data used to compute $D_0$ or tune hyperparameters? If so, this is leakage. Please rerun with target train only, and add at least one real cross domain benchmark (e.g., Office Home, DomainNet, VisDA, WILDS) with mean $\pm$ std over multiple seeds.

3. Please report the variable and constraint counts for the proposed linearization, and for the greedy ordering procedure, provide the per-iteration runtime and peak memory usage, along with any caching strategies employed.

4. The method is stratified, one sample per stratum, without replacement, and incorporates greedy in-cluster swaps. Please clearly state the underlying assumptions and provide either a formal proof or a bias bound for MMD/CORAL under this specific sampling procedure.

5. Please analyze the sensitivity of the final accuracy to $k$, $M$, $n_{\min}$, $\lambda$, and the bandwidth set, and provide the complete training configuration (batch size, learning rate schedule, weight decay). Also, correct the CORAL average delta inconsistency ($+7.5$ pp vs $70.5 - 62.0 = +8.5$ pp), clarify the overloaded notation for $k$, and validate the results on additional backbones (e.g., ResNet50, ViT-B/16, or ConvNeXt-T).

---

### Official Review · Reviewer_vt31 · 2025-11-03

**Soundness:** 2
**Presentation:** 2
**Contribution:** 2
**Rating:** 2
**Confidence:** 4

**Summary:**

The paper builds upon [Anonymous 2025] which proposed stratified sampling (as opposed to random sampling) of mini-batches during training to reduce the variance in estimation of domain discrepancy that arises when training with mini-batches.

**Strengths:**

Extending the idea of reducing the variance in domain alignment loss by introducing order of sampling is novel.

**Weaknesses:**

1. What is the difference between the terms L^m_disc and \hat{L}m_disc - These terms are not defined. If we assume they are the 2nd term in the objective (1) for the minibatch (m), what then is the difference between \hat{} and no \hat{}. This is regarding the lines 117 - 118.
2. How is D_0 even estimated? Wouldn’t this require using all training data in the source and target. This is not feasible for large datasets.
3. Lack of Novelty. The proposed approach is different from Anonymous 2025 through the idea of ORDERED. But, it is not clear why ORDERED is even necessary. Why is it necessary or useful to align \hat{D}^m with D_0 in Eq. 6. What is the intuition behind this regression based loss. \hat{D}^m is the distance estimated with a subset of samples while D_0 is estimated with all the samples. Just because they have the same scalar value, how does that ensure anything?
4. Inadequate experiments. There is insufficient empirical evidence to show the proposed method is of value to the DA community. There is only one dataset used for evaluation. It is not a standard dataset used by the DA community. There is no comparison to any baselines. There is little context to Table 1 and the numbers there do not help in understanding the proposed model.
5. Tedious notation. The reviewer spent quite some time trying to give the benefit of doubt to the authors. Section 2.4 was difficult to follow starting with lines 213 owing to cryptic or inadequate explanation for some of the terms.

**Questions:**

None.

---

### Note · Authors · 2025-11-24

I have read and agree with the venue's withdrawal policy on behalf of myself and my co-authors.